# Lamprey Wound Healing and Regenerative Effects: The Collaborative Efforts of Diverse Drivers

**DOI:** 10.3390/ijms24043213

**Published:** 2023-02-06

**Authors:** Shushen Li, Zhiyuan Zhao, Qingwei Li, Jun Li, Yue Pang

**Affiliations:** 1College of Life Sciences, Liaoning Normal University, Dalian 116081, China; 2Lamprey Research Center, Liaoning Normal University, Dalian 116081, China

**Keywords:** lamprey, skin wound, regenerate, lipolysis

## Abstract

Skin is a natural barrier between the body and the external environment, and this important multifunctional organ plays roles in body temperature regulation, sensory stimulation, mucus secretion, metabolite excretion and immune defense. Lampreys, as ancient vertebrates, rarely experience infection of damaged skin during farming and efficiently promote skin wound healing. However, the mechanism underlying these wound healing and regenerative effects is unclear. Our histology and transcriptomics results demonstrate that lampreys regenerate a nearly complete skin structure in damaged epidermis, including the secretory glands, and will almost not be infected, even if experiencing full-thickness damage. In addition, *ATGL*, *DGL* and *MGL* participate in the lipolysis process to provide space for infiltrating cells. A large number of red blood cells migrate to the site of injury and exert proinflammatory effects, upregulating the expression of proinflammatory factors such as *IL*-*8* and *IL*-*17*. Based on a lamprey skin damage healing model, adipocytes and red blood cells in the subcutaneous fat layer can promote wound healing, which provides a new approach for the study of skin healing mechanisms. Transcriptome data reveal that mechanical signal transduction pathways are mainly regulated by focal adhesion kinase and that the actin cytoskeleton plays an important role in the healing of lamprey skin injuries. We identified RAC1 as a key regulatory gene that is necessary and partially sufficient for wound regeneration. Insights into the mechanisms of lamprey skin injury and healing will provide a theoretical basis for overcoming the challenges associated with chronic healing and scar healing in the clinic.

## 1. Introduction

Skin is a stratified squamous epithelial organ and the primary barrier to the external environment in all animals [1]. The skin is very important for the maintenance of physiological functions, such as preventing fluid loss, stabilizing body temperature, transmitting sensory input [2] and immune defense [3]. The skin can usually be divided into three layers, the epidermis, dermis and hypodermis [4], and shows some degree of specialization in fish, reptiles, birds and mammals. Most fish are covered with scales, and mucous cells in the epidermis secrete mucus, which provides an additional barrier to prevent pathogen access to subcutaneous tissues and the vascular system through the skin [5,6]. The skin of amphibians contains large numbers of mucinous and granulosa glands in the dermis [7]. Lampreys are among the oldest extant jawless vertebrates, and the skin structure of lampreys is relatively conserved and can also be divided into the epidermis, dermis and hypodermis. Ficalbi (1916) described three morphologically different epidermal cell types [8]. Mucous cells [9], granular cells [10] and skein cells [11,12] are the major cell types comprising the epidermis and are close to those found in fish. Unlike most fish, the outermost layer of the skin in lampreys is covered with a cuticle, which is smooth without scales [13]. Fibroblasts in the dermal layer as well as cells in other layers, such as white adipocytes in the subcutaneous fat layer, have only rarely been described.

Because the skin is constantly challenged by a wide variety of external factors, this organ is highly susceptible to trauma [14]. Damage to the skin must be promptly healed to restore its essential barrier function. Skin healing is a complex and dynamic biological process involving many different cell types, the extracellular matrix and mediators, such as neuropeptides [15], growth factors [16,17] and cytokines. In mammals, wound healing is a rapid and scar-free process and can achieve functional regeneration during the embryo stage [18], whereas in adults, scars form, and the associated process can be divided into three main stages: inflammation, proliferation and remodeling [19]. Skin damage, particularly chronic healing and infection, usually causes pathological scars and poses many hazards. Moreover, wound healing strategies applied in the clinic can achieve only pathological healing but not physiological regeneration [20] and require long-term treatment, which places a substantial economic burden on the global medical system. Therefore, studying the mechanism of healing after skin damage is very important.

Animal models have been developed to study the complex cellular and biochemical processes of wound healing and to evaluate the effectiveness and safety of potential therapeutic agents. Although wound healing research in mammalian systems, such as mice and pigs, has high medical relevance, it is costly, technically challenging and time-consuming [21]. The panniculus carnosus is found in the subcutaneous tissue of mice but is almost absent in humans. Wounds are mainly established by contraction and marginal healing instead of re-epithelization [22], and pig skin has a higher pH value and more subcutaneous fat [23]. Differences in anatomy and physiology are found among animal species, and no single model of skin wounds meets all the needs of researchers. Fortunately, the healing process is conserved in vertebrates. Therefore, the use of “lower” organisms is helpful for studying the mechanism of skin healing [24]. The skin structure of lampreys is conserved and divided into three layers: epidermis, dermis and hypodermis [9,10,11,12]. The skin contains functional cells such as mucous cells [9] and has no scales, hair follicles or other complex appendages. The skin of lampreys contains substances consistent with those found in other vertebrates, such as cholesterol esters, cholesterol, free fatty acids (FFAs), sulfated glycosaminoglycans [25] and a variety of collagens with triple helix regions [26], as well as many common molecules, such as IL-17 [27] and Ocln [28]. Earlier studies investigated *Flanobacterium phytophilum*, which causes skin lesions in lampreys [29], and the skin of allografted lampreys was also studied by histology and electron microscopy [30]. In the latest study, Ruixiang Sun et al. found that Lr-PGRN-S1 was highly expressed at the early stage of skin damage [31]. However, the study of the skin of primitive vertebrates such as lampreys remains limited, and few studies on the skin damage process in lampreys have been reported. To the best of our knowledge, the skin healing process of lampreys remains unknown. This study provides the first description of the process of skin healing in lampreys and the first identification of various types of cells and signaling pathways involved in the healing of skin damage, and the study thus provides a valuable model for determining the roles of adipocytes and red blood cells in skin regeneration and offers a new approach for improving regeneration in mammals. The results also confirm that lampreys can achieve functional regeneration after skin damage, which provides a new approach for the exploration of chronic healing and scar healing.

## 2. Results

### 2.1. Tissue Characteristics during Wound Healing after Lamprey Skin Damage

Electron microscopy [9,10,11,12] and conventional histology [14] results have shown that the skin structure of sea lamprey is relatively conserved, and the same is true of *Lethenteron reissneri*. However, to the best of our knowledge, the repair process remains unknown, and we gently scraped the lamprey epidermis to damage it. The lamprey skin tissue injury and sampling collection processes are shown in Figure 1A. To clarify the processes of lamprey skin injury and healing, we recorded the healing characteristics at different times after skin injury (Figure 1B). At the onset of injury (0 days after damage, Dam 0 d), the color of the adult lamprey dermis was nearly blue. In addition, the subcutaneous adipose layer with hyperemia was locally dilated with a small number of red blood cells (RBCs) and inflammatory cells, the melanocytes were gradually dispersed, and some adipocytes had disappeared. At 2 days after injury, the entire wound was congested, the subcutaneous fat layer was further expanded and filled with a large number of RBCs and granulocytes and melanocytes had migrated to the bottom of the hyperemic region. On the 7th day after injury, the subcutaneous fat layer of the wound was reduced, a small number of granular and skein cells appeared in the regenerated epidermis, and the migration of melanocytes was reduced. At 14 days after injury, the hyperemia had further subsided, the degree of cell differentiation in the regenerated epidermis was high, the melanocytes had migrated back to the dermis, the number of infiltrating cells was reduced, the thickness of the subcutaneous fat layer was significantly reduced and white adipocytes had reappeared. At 21 days, epidermal damage could still be observed, but the thickness of the epidermis was consistent with that of undamaged skin, no obvious boundary with the undamaged epidermis was detected and the organization was similar to that of undamaged skin (Figure 1C). We compared the processes of skin injury and healing between adult and larval lampreys, and the results showed that larval lampreys were more hyperemic (Appendix A). In addition, our results revealed that deep damage could delay the healing process (Appendix A).

To further define the healing process of the epidermis, serial sections of damaged skin at 7 d were prepared from wax blocks and sorted 1–6 from the wound edge to the wound center (Figure 1D). The results showed that the average thickness of the epidermis was smaller closer to the wound center and that the epidermis was thicker farther away from the center of the wound bed. The epidermal thickness increased from 20 to 40 μm, but the differentiation degree of granular cells and skein cells was independent of the distance from the wound center (Figure 1E). The epithelial cells at the undamaged site underwent changes in the cytoskeleton and cell morphology, which were characterized by anterior and posterior polarity replacing upper and lower polarity, and cells migrated to the wound bed from the damaged site to fill the wound (Figure 1F).

### 2.2. Functional Regeneration of Damaged Skin Tissue

We verified that the newly formed epidermis is consistent with undamaged skin in terms of morphology and cell composition, but whether they can still function properly is unknown. Mucus secretion is an important function of the epidermis, and collagen deposition is also important for skin function. To determine the functional recovery ability of lamprey skin 21 days after epidermal injury, functional healing was detected by Masson’s trichrome staining and PAS-AB staining. Collagen deposition usually occurs when the skin of mice or other model organisms is damaged, and this deposition affects the mechanical strength of wound healing and may also lead to defects in wound healing [32]. Rita et al. described the structural and morphological characteristics of the regenerated skin of sea bream within 4 days after injury by Masson’s trichrome staining [33]. However, no corresponding description of lamprey is available. Our study showed no significant collagen deposition within 7 days of skin injury. At 14 days after injury, collagen deposition in the muscle layer had increased significantly, and at 21 days after injury, the increased collagen fibers were again replaced by muscle fibers as the damaged skin was repaired (Figure 2A). In other words, collagen deposition also occurs during the process of lamprey skin injury but disappears as healing progresses. PAS-AB staining is often used to identify the mucus secretion of mucous cells. The PAS-AB staining of *Lethenteron reissneri* is basically consistent with that of sea lamprey [14]. The PAS-AB staining results showed that the cuticle was characterized by a strong signal, granular cells exhibited a weak PAS-positive signal, the basal and adipose layers showed a red PAS-positive signal, mucous cells close to the epidermis presented a double-positive purplish red signal and epidermal damage did not change the secretory patterns of other tissues. During the progression of epidermal healing, the regenerating epidermis retained its secretory function, although a diminished blue positive signal was observed in the cuticle (Figure 2B). At this time point, the melanin deposition function of the wounds was not effectively healed, but gradual repair was detected in the long term (Figure 2C). However, we observed that lampreys could achieve nearly complete wound healing of epidermal structures, including epidermal appendages such as mucous cells and granule cells (Figure 2D).

### 2.3. The Collaborative Efforts of Diverse Drivers Induce Cell Migration in Response to Epidermal Damage

We sampled damaged skin without muscle tissue, compared the damaged and control groups and categorized genes as differentially expressed genes (DEGs) if |log2-fold change| ≥ 1.5 (*p* < 0.05). The distribution trends of the DEGs were mapped in a volcano plot. In comparison with the levels in the control group, skin damage resulted in 345 upregulated genes and 490 downregulated genes at Dam 0 d, 638 upregulated genes and 509 downregulated genes at Dam 7 d, 688 upregulated genes and 651 downregulated genes at Dam 14 d, and 686 upregulated genes and 715 downregulated genes at Dam 21 d (Figure 3A and Appendix A). More DEGs were detected at the late stage of damage than at the early stage of damage. The results from a cluster expression analysis showed that the expression of DEGs varied significantly in the damaged skin among different time points, which suggests that the primary event changed during these periods (Figure 3B), consistent with the results shown in Figure 1B,C. A GO enrichment analysis of DEGs was carried out to explore the biological functions of these DEGs at each healing stage. Gene Ontology (GO) is a comprehensive database describing gene functions, which can be divided into three categories: biological process (BP), cellular component (CC) and molecular function (MF) (Figure 3C and Appendix A). The GO annotation provided interesting information regarding BPs. Genes related to the extracellular matrix (ECM) and actin were highly expressed throughout the healing stage and were closely related to cell migration and differentiation. ECM components create “scaffolding”, which stimulates cell adhesion and migration and mediates interactions between cells and between cells and ECM proteins at the early stage of inflammation; in addition, at the late stage of proliferation and remodeling, wound healing is regulated and actin responds to changes in the ECM to cause cell migration. These effects enable multiple types of cells to accumulate at the wound site to participate in healing. In contrast, differences are found between the early stage (Dam 0 d-Dam 7 d) and the middle and late stages. Genes related to oxygen binding were upregulated at the early stage, whereas those related to DNA replication and the cell cycle were upregulated at the middle and late stages (Dam 14 d-Dam 21 d), indicating that early cells respond to damage by increasing oxygen binding and that a large amount of cell proliferation may occur at the middle and late stages of damage. In addition, a total of 9 (Dam 0 d), 15 (Dam 7 d), 12 (Dam 14 d) and 9 (Dam 21 d) terms with *p* values < 0.05 were considered significantly overrepresented. Twelve of the most prominent GO terms were selected. These DEGs respond to damage, and changes in the ECM drive the process.

To identify the main pathways that may be involved at each stage of lamprey epidermal damage, a KEGG analysis was conducted (Figure 3D and Appendix A). The upregulated signaling pathways were mainly ECM receptor interaction (dre04512), focal adhesion kinase (dre04510) and regulation of the actin cytoskeleton (dre04810). All types of cell surface adhesion complexes are involved in cytoskeletal organization and signaling pathways. Signaling pathways related to DNA replication were also enriched 14 days after damage. These results are consistent with results from the GO enrichment analysis.

In our study, DEGs were annotated as cytoskeleton-related, adhesion-related and ECM-related KEGG signaling pathways, which revealed the enrichment of lamprey DEGs associated with epidermal damage. Many identified genes were significantly upregulated, including *ARP2/3*, *COL2A1*, *DOCK1*, *FAK1*, *FGF9*, *Fgf4a*, *ITGA11*, *ITGA8*, *PDGFRA*, *RRAS2*, *RAC1*, *RHOA* and *VAVA3*, and these genes ultimately affected changes in F-actin. A common molecule, *Rac1*, which is located downstream of the two signaling pathways of actin skeleton regulation and focal adhesion kinase, was identified (Figure 3E). Lamprey Rac1 has high homology and similarity with mouse Rac1, and we inhibited its expression using Ehop-016. The results showed that hyperemia was relieved after 2 days of damage in the high-concentration and low-concentration Rac1 inhibition groups (Figure 3F), and this change disappeared at Dam 7 d (Appendix A).

Quantitative real-time PCR was performed to confirm the expression of genes identified by Illumina sequencing analysis (Figure 3G). The qPCR results were consistent with the transcriptome analysis results. However, *PDGFRA* experienced transient upregulation at Dam 2 d, which was subsequently consistent with the transcriptome data. After epidermal damage in lampreys, some small G proteins in the focal adhesion kinase signaling pathway, such as *Rac1*, and tyrosine kinases, such as FAK, were significantly upregulated, *ITGA8* was significantly upregulated and *ITGA11* was significantly downregulated. We also characterized the functions of these proteins (Table 1). These results provide some support for the RNA-seq results, indicating that the above mentioned transcriptome results have high reliability (Figure 3G).

### 2.4. The Hypodermis Is the Main Site of Healing of Epidermal Damage

To further search for the key genes involved in the healing of damaged skin in lampreys, we targeted the hypodermis early after epidermal damage. We collected samples from only the hypodermis and performed RNA-seq (Figure 4A) and H&E staining. The main cell types in the hypodermis after damage were RBCs, leukocytes and neutrophils (Appendix A). Based on the RNA-seq results, we compared the damaged and control groups and categorized genes as DEGs if |log2-fold change| ≥ 2 (*p* < 0.05). The analysis identified 3485 and 2671 upregulated genes and 2413 and 2407 downregulated genes in the hypodermis on days 1 and 2 after epidermal damage, respectively (Figure 4B). A cluster expression analysis showed that the DEGs were similar between Dam 1 d and Dam 2 d (Figure 4C). Further GO analysis revealed that these DEGs could be classified into 861 terms and 850 terms, and most of these terms were BP terms (52.6% and 53.9%, respectively). Among the identified terms, the terms with the greatest enrichment were the following: CC, extracellular region (GO:0005576, 12.1%, 11.6%); BP, transmembrane transport (GO:0055085, 10.8%, 9.3%); and MF, transporter activity (GO:0005215, 7.5%, 6.9%). Many terms related to cell adhesion (GO:0007155) and cytoskeleton (GO:0005856) were also found (Figure 4D). The KEGG analysis revealed 150 and 148 signaling pathways in total, and among these, the signaling pathways with the most genes included focal adhesion and the ECM-receptor interaction signaling pathway (Figure 4E). However, each term contained more genes than the above mentioned transcriptome. In addition, some genes related to the calcium regulation pathway were upregulated, indicating that calcium plays a regulatory role in early inflammation and cell migration after damage. Several important genes selected above were upregulated, but *PDGFRA* and *ITGA11* were not. In addition, the hypodermis showed higher expression of *COL4A2* (Figure 4F).

### 2.5. Adipocytes and RBCs Play Vital Roles in Promoting Wound Healing

In mice, skin adipocytes undergo lipolysis after skin injury, and the ablation of dermal adipocytes reduces blood circulation reconstruction [47], indicating that adipocytes play an important role in skin repair. We found that adipocytes also participate in epidermal damage repair in lampreys. According to the histochemical results, white adipocytes gradually decreased in number or even disappeared during the repair process. At the completion of healing, the lost cells had regenerated (Figure 5A). We screened several DEGs belonging to the lipase family, including *ATGL*, *DGL*, *MGL*, *PNLIPRP2* and *BAL*, and detected their mRNA expression levels by RT–PCR. The results showed that during the progression of lipolysis, the expression of these genes was significantly upregulated (*p* < 0.05) (Figure 5B).

As mentioned above, an increasing number of RBCs had infiltrated the hypodermis 48 h after damage (Figure 5A). To further validate the function of RBCs, we examined the expression levels of *IL-17*, *IL-8* and *IL-1β-like* mRNAs, which showed upregulation of proinflammatory factors in RBCs (Figure 5C). Regarding hypodermis cell proliferation, our results suggest that these cells can increase in number by proliferating in addition to infiltrating from other sites (Figure 5D).

## 3. Discussion

The skin repair process of fish, including zebrafish [21] and sea bream [33], and higher mammals involves inflammation, proliferation and remodeling. The skin healing process in lampreys is conserved and involves these processes. After mammalian skin damage, blood coagulation occurs first, providing chemokines to attract inflammatory leukocytes and extracellular matrix proteins to form the matrix of migrating cells [48]. However, at the early stage of lamprey skin damage, no formation of an external fibrin clot is observed, which is similar to the healing process of zebrafish [21]. The inflammatory stage occurs as rapidly as observed in fish and humans. After damage, neutrophils are the first to reach the wound in fish and mammals [49], but in lampreys, RBCs also appear to migrate very rapidly to wounds. The mast cell release of proinflammatory and immunomodulatory mediators has been well documented. Our results show differential expression of the tryptase gene, which indicates that mast cells may be involved in the healing of lamprey skin after damage [50]. Granulation tissue formation and vascularization occur after inflammation [51]. Although these effects were not observed in lampreys, we detected the expression of some angiogenesis-promoting genes, such as Ang-2 [52]. Rodents heal mainly by contraction, whereas in humans, re-epithelialization accounts for up to 80% of wound closure [53]. Wounds in lampreys seal through the process of re-epithelization, and the re-epithelization process is faster than that of human beings but slower than that of zebrafish. At present, our focus is limited to epidermal damage, and more signaling pathways and molecules related to healing may be found for full-thickness skin damage.

In recent years, the nonenergy storage functions of dermal white adipose tissue, such as wound healing [47,54] and hair follicle formation [55], have been studied, and in mice, these functions have been found to be involved in the repair of skin damage [50]. We also found such functions in lamprey (Figure 5D). PNLIPRP2 is a member of the classic triglyceride lipase family, which cleaves neutral and charged fats [56]. Therefore, we speculate that the metabolism of neutral and charged fats is involved in the healing of skin damage. We also speculate that lipolysis produces eicosanoids, and their further metabolites can effectively participate in disease, proliferation, migration, angiogenesis and remodeling. DGL produces the endocannabinoid 2-arachidonoylglycerol, and MGL then hydrolyzes 2-arachidonoylglycerol, which is a potent ligand within the endocannabinoid system, into arachidonic acid, a precursor for prostaglandin synthesis [57]. Further research should verify the specific role of lipolysis products. During the process of skin damage healing in lampreys, multiple lipolytic enzyme genes play a role, and these not only provide space but may also participate in the wound healing response.

In lampreys, RBCs contain nuclei. Although mammalian RBCs have no nucleus, research shows that RBCs are a key contributor to innate immunity. Nonmammalian RBCs directly participate in the immune response [58], and Atlantic salmon (*Salmo salar* L.) erythrocytes increase the transcription of antigen-presenting and viral-response genes during *Piscine orthoreovirus* (PRV) infection and upregulate interferon alpha (IFN-α) in response to infectious salmon anemia virus (ISAV), contributing to host resistance [59,60]; the same has been observed in lampreys. To eliminate the role of a small amount of inflammation, we stimulated lampreys with *Vibrio anguillarum* and extracted whole blood to facilitate the separation of RBCs and thus prove that RBCs promote inflammation (Appendix A).

The dynamics of the actin cytoskeleton can directly influence cutaneous stem cell behaviors, including wound repair [61]. In addition, actin cytoskeleton remodeling promotes cell migration in skin injury [62]. Our results revealed that regulation of the actin cytoskeleton (dre04810) and ECM-receptor interaction (dre04512) were significantly enriched. The dynamic regulation of the F-actin cytoskeleton is essential for many physical cellular processes, including cell adhesion, migration and division. Each process requires precise regulation of the cell shape and the generation of mechanical forces [62]. We speculate that the epidermal damage healing process in lampreys requires a large amount of cell migration and cell adhesion, which are also reflected in the enrichment of the FAK signaling pathway because this pathway is involved in focal adhesion [63]. These main DEGs may play such a role. The ECM is a fundamental component of multicellular organisms and plays vital roles in providing structure, guiding the migration and polarity of cells and maintaining the morphogenesis and coherence of tissues [64]. Here, we focus on the deposition of collagen. Among mammals, granulation tissue is largely composed of collagen III, which is partially replaced by the stronger collagen I as remodeling of the wound progresses [65]. In lampreys, our results show that type II collagen was upregulated at the early stage of skin damage. However, in adipose tissue, type IV collagen was significantly upregulated.

We found that external stimuli that control cell migration are transduced into intracellular biochemical signals through the interactions of transmembrane integrins that bind to ECM proteins or mechanical stimuli to promote deformation of the actin cytoskeleton, causing a large number of cells to infiltrate the wound bed such that the proliferation and remodeling stage can be completed. To comprehensively describe this fascinating biological process during wound healing, easily standardized experimental mice, zebrafish and other organisms are often used as skin healing models, but no single skin damage model can meet all the requirements of researchers. The skin structure is conserved among vertebrates, similar to that in lampreys. Our experimental lampreys exhibited almost no infection due to skin damage, which shows that lampreys have advantages in challenges to the skin barrier. Unlike that of fish, the lamprey epidermis has a cuticle, which is the outermost layer of human skin [66]. Thus, lampreys are more suitable for skin damage research than fish. The epidermis of lampreys is mainly composed of mucous cells, and mucus performs a variety of functions to prevent mechanical damage and limit the invasion of pathogens [67]. Therefore, the skin is more similar to a skin accessory organ in lamprey due to its simplicity and few cutaneous appendages, such as squamous hair follicles (Figure 1D), which indicates that lamprey can provide a simplified skin healing model. Skin wound healing has a fascinating mechanism and represents an evolutionary advantage for mammals [48], and our study confirms this view.

The regeneration mechanism of skin appendages is a key issue. Our results indicate that the skin appendages of lampreys can achieve functional regeneration, the cell type and tissue structure of the new epidermis are almost the same as those of undamaged skin (Figure 1D) and monoglandular epidermal cells can play a role in newborn skin (Figure 2); however, this still needs further verification, such as the activity of the main components in the mucus. Although the wound can still be seen at Dam 21 d, we believe that this is caused by the uneven distribution of melanocytes, which can promote healing for a long time (Figure 2D). Another interesting result is that the regeneration ability of lampreys did not decrease with growth. This finding is promising, and it is currently believed that engrailed-1 lineage fibroblasts promote scar healing in wounds [68]; the differential expression of these genes was not detected after skin damage in lampreys. These findings provide new insights into the study of scar-free healing.

## 4. Materials and Methods

### 4.1. Experimental Design and Sampling Procedure

Adult lampreys (*Lethenteron reissneri*, body length: 15 cm ± 1 cm) were obtained from Huanren Manchu Autonomous County, Benxi, Liaoning Province, China. The lampreys were then maintained in an aquarium at 4 °C. All lamprey handling and experimental procedures were approved by the Animal Welfare and Research Ethics Committee of the Institute of Dalian Medical University. For the epidermal regeneration challenge, adult lampreys (N = 30) were divided randomly into five groups: no wound (CTL), 0 days after damage (Dam 0 d), 2 days after damage (Dam 2 d), 7 days after damage (Dam 7 d), 14 days after damage (Dam 14 d), and 21 days after damage (Dam 21 d). After anesthetization with 0.05% MS-222 (tricaine methanesulfonate, Sigma–Aldrich, E10521, St. Louis, MO, USA), the body surface was scraped using a sterilized scalpel to create a wound bed (1.5 cm × 0.5 cm) until no more mucoid tissue could be scraped. The whole process was performed gently to minimize damage to the dermis. For full-thickness damage, a scalpel with only a 5-mm blade exposed was used to make a 0.5-cm-long wound on the side of the lamprey parallel to the body. Images were captured with a zoom stereo microscope (SMZ1500, Nikon, Japan).

### 4.2. Histology of Lamprey Skin

Intact and regenerating skin samples were harvested, fixed in Bouin’s solution and embedded in paraffin wax. The paraffin-embedded tissues were sectioned to a thickness of approximately 4 µm and stained with hematoxylin and eosin (H&E), Masson’s trichrome (Sangon Biotech, Shanghai, China) [33] and PAS (Periodic Acid Schiff, Solarbio, G1281, Beijing, China)-AB (Alcian blue, Sangon Biotech, Shanghai, China) [69]. In brief, the slices were hydrated with xylene and a series of graded ethanol solutions and stained (aniline blue instead of light green). For the sampling of subcutaneous adipose tissue, the dermis was removed, and the muscle layer was removed after 1 or 2 days of epidermal damage in juvenile lamprey and stained using the method described above. The integrity of the skin structure and the types of cells were examined under a microscope.

### 4.3. RNA Extraction, cDNA Synthesis, Library PREPARATION, Sequencing and Transcriptome Assembly

Total RNA from unwounded and epidermis-damaged tissues (Dam 0 d, Dam 7 d, Dam 14 d and Dam 21 d) was harvested with TRIzol reagent. The method used to inflict damage was the same as that described above, and the muscle layer was removed, assessed with the RNA Nano 6000 Assay Kit for the Bioanalyzer 2100 system and treated with DNase I. Reverse transcription was performed. Blood and hypodermis samples from the damaged area were collected and treated using the same method for transcriptome data, as previously reported [70]. In brief, first-strand cDNA and second-strand cDNA were synthesized using DNA polymerase I. The RNA sequence library was constructed after hybridization, screening, amplification and purification, and its quality was evaluated with an Agilent Bioanalyzer 2100 system.

### 4.4. Identification of DEGs and GO and KEGG Enrichment Analyses

The FPKM of each gene was calculated based on the length of the gene and the read count mapped to the gene. Prior to differential gene expression analysis, for each sequenced library, the read counts were adjusted with the edgeR program package using one normalized scaling factor, and genes were classified as significantly differentially expressed if *p* value < 0.05 and log2-fold change ≥ 1.5. GO enrichment analysis and KEGG enrichment analysis of DEGs were performed with Cluster Profiler (3.4.4) software.

### 4.5. Quantitative Expression Analysis (qPCR)

Unwounded and epidermis-damaged tissues (skin samples without the muscle layer and hypodermis samples) were obtained using the above-described methods, and total RNA was extracted and treated with DNase I (TaKaRa, Beijing China). qPCR was performed as previously described [71]. The primer sequences and accession numbers of the genes are provided in Appendix A. Each reaction was performed in triplicate, and the data were normalized to the level of L-gapdh as an internal control. The differences in gene expression values were analyzed by Student’s *t* test. Differences were considered statistically significant if *p* < 0.05.

### 4.6. Pharmacological Inhibition with the Rac1 Inhibitor Ehop-016 Impairs Epidermal Damage in Lampreys

Lampreys treated with the Rac1 inhibitor were cultured using the same above-described approach. The working solution was prepared using DMSO, PEG300 and Tween-80 according to the manufacturer’s instructions (Shanghai Yuanye Biological, Shanghai, China). Subsequently, 1.8 mL of physiological salt solution was added, and the samples were stored at −20 °C. The above-described method was used to damage the epidermis of the lampreys, and before damage, the lampreys were injected with the inhibitor before damage at a low (15 mg/kg) or high concentration (50 mg/kg) or with the carrier control (DMSO, PEG300, Tween-80); the inhibitor was injected subcutaneously every other day. The damaged tissues were collected after 2 and 7 days for staining.

### 4.7. Cell Proliferation Assay

EdU-injected lampreys were cultured under the same conditions, and 50 mg/kg EdU (Beyotime, Shanghai, China) was injected intraperitoneally at the indicated time points and detected according to the manufacturer’s protocols. After another injection 24 h later, at Dam 7 d, samples were collected and stained. EdU incorporation was evaluated using the Click-it EdU imaging technique.

### 4.8. Statistical Analysis

In this study, GraphPad Prism 8.0 was used to analyze and plot the experimental results. The experimental results were independently verified three times. The values are expressed as the means ± standard deviations (x ± SDs). The variance was calculated by *t* test: *p* < 0.05 indicated a statistical significance, and *p* < 0.01 indicated an extremely significant difference (* *p* < 0.05, ** *p* < 0.01).

## 5. Conclusions

Overall, in this study, we demonstrated that epidermal damage in lampreys can induce functional regeneration, and almost no difference was observed between newly formed and undamaged epidermis. These results suggest that lamprey is a potential simplified model for skin repair. We also found that many types of cells involved in the regulation of the actin cytoskeleton (dre04810) and focal adhesion kinase (dre04510) migrate to the wound to promote repair, which provides a pool of repair-promoting genes. We further verified that RBCs and adipocytes participate in the skin repair process through proinflammatory and lipolysis, respectively. Our data open the door to the development of new skin damage models and provide a basis for further exploration of repair-promoting genes and strategies found in lampreys, a primitive vertebrate.

## Figures and Tables

**Figure 1 ijms-24-03213-f001:**
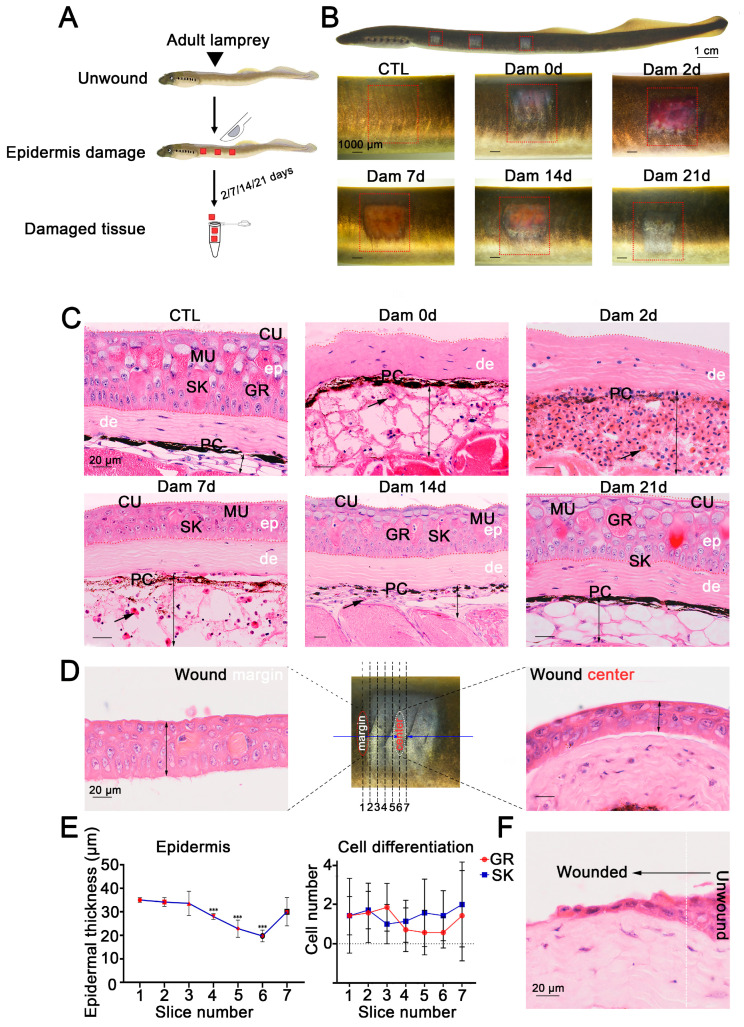
Wound healing process of the damaged lamprey (*Lethenteron reissneri*) epidermis. (**A**) Five time points were selected to characterize the healing process of lamprey. (**B**) Representative images of the skin healing process of adult lamprey after epidermal damage; three wounds on each side are shown (red square). (**C**) Representative images of H&E-stained sections of the wound bed. Significant changes are visible in the subcutaneous fat layer, and massive red blood cell infiltration was detected (black arrow). Scale bars, 20 μm. (**D**) Serial sections were prepared from the wound edge to the center. (**E**) Statistical analysis of the numbers of granular cells and skein cells (left) and the epidermal thickness of sections 1 to 7. *** *p* < 0.001 versus slice 1. (**F**) Migration of epidermal basal cells to the wound bed. ep: epidermis; de: dermis; PC: pigment cell; MU: mucous cell; SK: skein cell; GR: granular cell; CU: cuticle.

**Figure 2 ijms-24-03213-f002:**
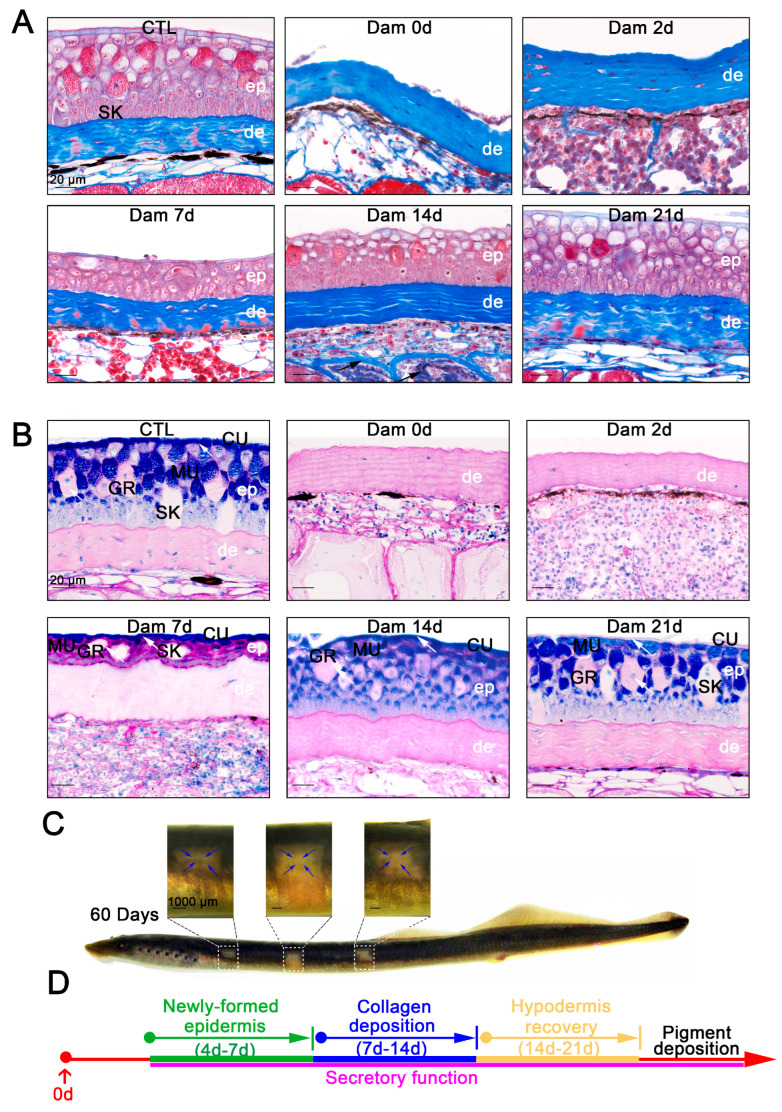
Histological staining results and long-term healing of skin damage in lampreys. (**A**) Masson’s trichrome staining results of skin tissue from the unwounded group until 21 days after damage. Scale bars, 20 μm. (**B**) PAS-SB (Alcian blue-periodic acid Schiff staining) results for the undamaged group until 21 days after damage; acidic mucus is stained blue, neutral mucus is stained red, mixed mucus is stained blue purple and the nucleus is stained light blue. The new epidermis has a secretory function (white arrow). Scale bars, 20 μm. (**C**) Pigment deposition in the wounds of lamprey 60 days after damage. Pigment deposits from the edge of the wound to the center. Scale bars, 1 cm. (**D**) Major events in the healing process.

**Figure 3 ijms-24-03213-f003:**
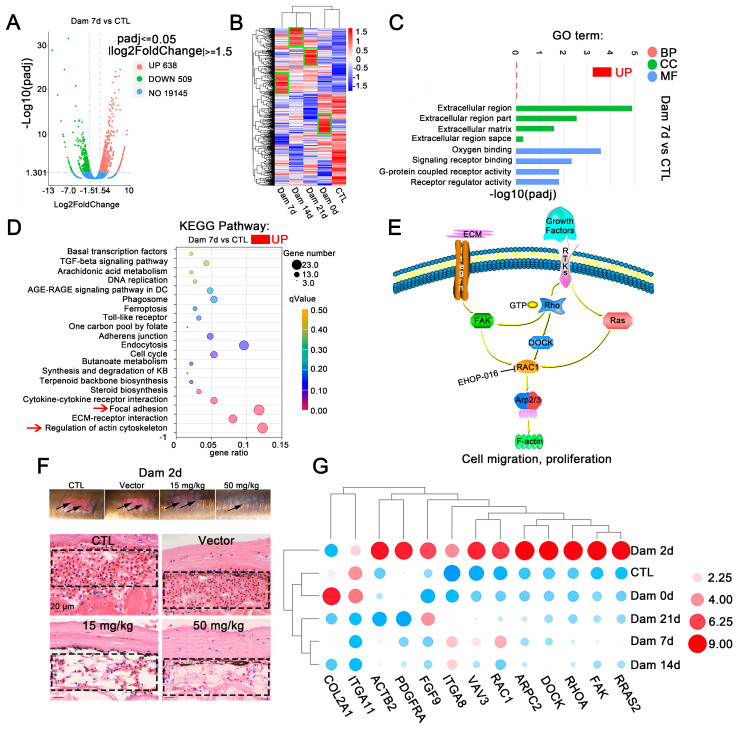
RNA-seq results of lamprey epidermis and dermis tissue after skin damage. (**A**) Volcano plot of the differentially expressed genes (DEGs) after skin damage in lampreys. (**B**) In the cluster expression analysis, the DEGs changed at different time points (green box). (**C**) GO functional category analysis. (**D**) KEGG pathway classification analysis and the DEGs involved in the main pathway; the red arrow indicates major signaling pathways. (**E**) The main DEGs eventually induced high expression of F-actin, changed the cytoskeleton, responded to mechanical injury and induced cell migration. (**F**) Early effect of the addition of the Ehop-016 inhibitor. Scale bars, 20 μm. (**G**) Thirteen genes related to regulation of the actin cytoskeleton and focal adhesion kinase were associated with significant changes in two signaling pathways related to focal adhesion kinase. DC: diabetic complication; KB: ketone bodies.

**Figure 4 ijms-24-03213-f004:**
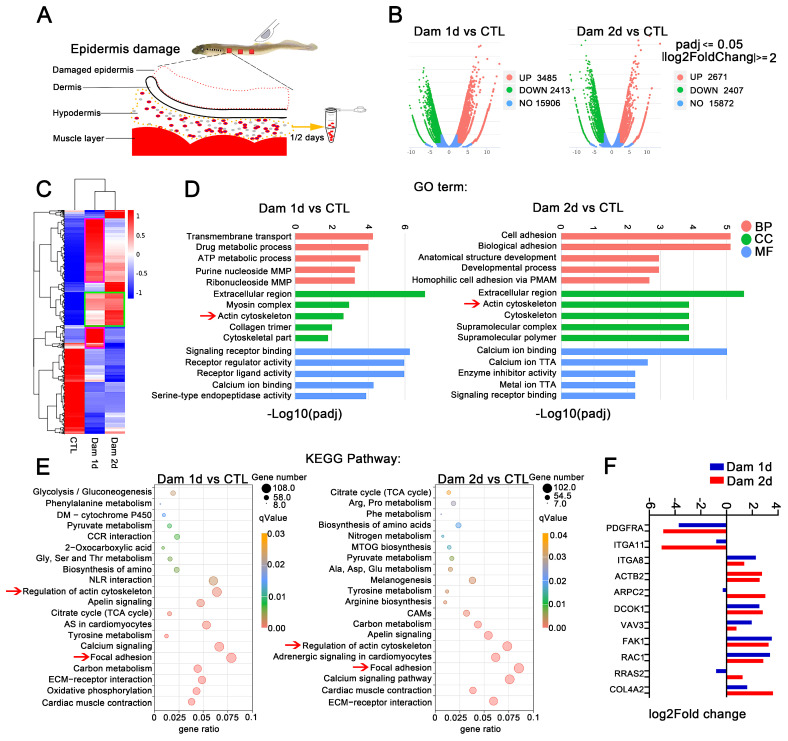
RNA-seq analysis of the hypodermis after epidermal damage. (**A**) Schematic diagram of hypodermis transcriptome sampling after epidermal damage. (**B**) Volcano plot of the differentially expressed genes (DEGs) in lampreys after skin damage. (**C**) Cluster expression analysis. Shared DEGs (green box) and unique DEGs (purple box) were found at the two time points. (**D**) GO functional category analysis. (**E**) KEGG pathway classification analysis; the red arrow indicates major signaling pathways. (**F**) Expression of selected genes in the subcutaneous fat layer. MMP: monophosphate metabolic process; PMAP: plasma membrane adhesion protein; TTA: transmembrane transporter activity; DM: drug metabolism; CCR: cytokine−cytokine receptor; NLR: neuroactive ligand−receptor; AS: adrenergic signaling; CAMs: cell adhesion molecules; MTOG: mucin type O−glycan. The yellow arrow means to receive samples one day and two days later. The red arrow is to better notice the key signal pathway.

**Figure 5 ijms-24-03213-f005:**
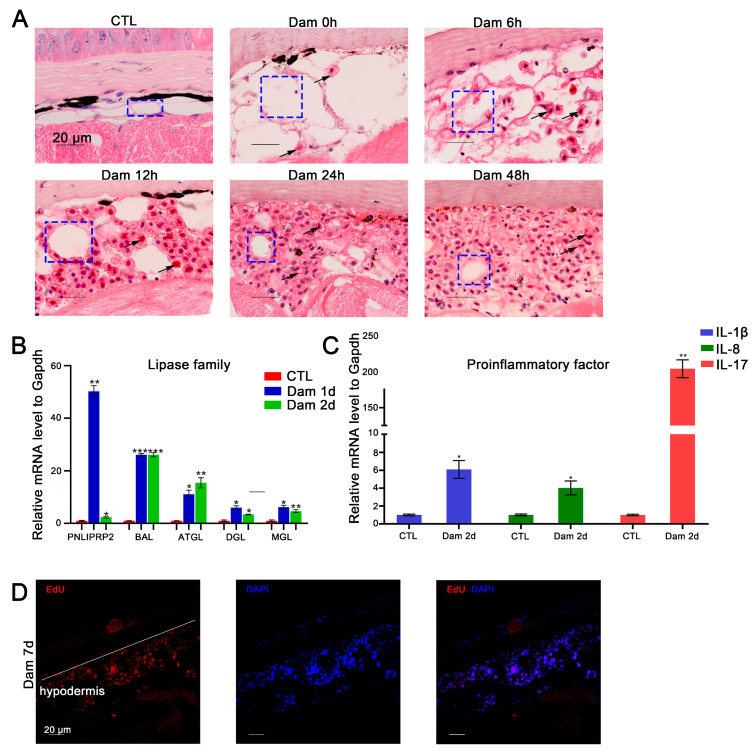
Functional analysis of red blood cells (RBCs) and adipocytes after epidermal damage. (**A**) H&E staining of RBC infiltration in the subcutaneous adipose layer after damage; lipolysis of adipocytes (blue box), gradual infiltration of RBCs (black arrow). Scale bars, 20 μm. (**B**) The significantly changed lipase genes in the RNA-seq data were verified by RT–PCR. (**C**) Results of RT–PCR analysis of proinflammatory factors in RBCs. * *p* < 0.05 versus CTL, ** *p* < 0.01versus CTL, *** *p* < 0.001 versus CTL. (**D**) EdU proliferation staining results of infiltrating cells in the subcutaneous fat layer. Scale bars, 20 μm.

**Table 1 ijms-24-03213-t001:** Identification of the molecule relative to wound healing in lamprey.

Gene Symbol	Description	ORF (bp)	Amino Acids	Function	References
ARP2/3	actin related protein 2/3 complex	654	217	It induce an explosive actin polymerization in response to signaling pathways	Molinie et al. (2018) [34]
ACTB2	actin, beta 2	1131	376	It supports fundamental cellular processes including cell growth and migration, cell morphology, cytoskeletal, cytokinesis and maintenance of cell polarity	Atzmony et al. (2020) [35]
COL2A1	collagen type II alpha 1 chain	1176	391	It forms a covalently cross-linked fibrillar network in the extracellular matrix that provides tensile strength to connective tissues, acts as an autocrine factor of proliferation and differentiation via multiple downstream effectors	Zhang et al. (2020) [36]
DOCK	dreadlocks	5733	1910	It plays a role in key cellular processes such as cell migration and adhesion	Benson et al. (2021) [37]
FAK1	focal adhesion kinase 1-like	3639	1212	It recognised as a hub in the interactome of focal adhesions, a protein complex at the centre of mesenchymal cell migration	Horton et al. (2015) [38]
FGF9	fibroblast growth factor 9	567	188	It can regulate tissue healing, cell proliferation, differentiation and cell migration	Huang et al. (2021) [39]
ITGA11	integrin subunit alpha 11	489	162	It binds with high affinity to collagen I and and is involved in myofibroblast differentiation and collagen reorganization	Grella et al. (2016) [40]
ITGA8	integrin subunit alpha 8	2775	924	It regulates cell survival, proliferation, differentiation, and migration	Herdl et al. (2017) [41]
PDGFRA	platelet derived growth factor receptor alpha	4878	1625	It accumulation of myofibroblasts and matrix deposition, leading to stimulation of cell growth, differentiation, and migration, promote adipose tissue fibrosis	Song et al. (2020) [42] Andrae et al. (2008) [43]
RRAS2	RAS related 2	534	177	It regulates cell migration, survival, inside-out integrin signaling, platelet aggregation, hemostasis and thrombosis	Janapati et al. (2018) [44]
RAC1	Rac family small GTPase 1	579	192	It induce the formation of membrane protrusions and retractions as well as the regulation of actin polymerisation into filaments	Nguyen et al. (2018) [45]
RHOA	ras homolog family member A	582	193	It established as crucial signaling effectors regulating cellular morphology and, consequently, locomotion	Nguyen et al. (2018) [45]
VAVA3	vav guanine nucleotide exchange factor 3	2775	924	It regulate cytoskeleton remodeling associated to barrier stability, integrin-dependent adhesion, spreading, polarity, and phagocytic functions	Badaoui et al. (2020) [46]

## Data Availability

The data supporting the results of this study are included in this paper. The data obtained in this study and the materials used in this study can be obtained from the corresponding authors according to reasonable requirements.

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
