# Peer review of "Lamprey Wound Healing and Regenerative Effects: The Collaborative Efforts of Diverse Drivers"

_ijms, 2023, doi:10.3390/ijms24043213_

Round 1

Reviewer 1 Report

The manuscript describes the wound healing process in lampreys. The results are well presented and discussed in detail. A minor spelling check is required.

Here are a few questions and suggestions:

1. Page #2, Can you explain " Fortunately, the healing process is conserved in vertebrates"?

2. Page #2, Line 9 (from top), correct "four" to three.

3. How did you decide to run the study for only 21 days? Why not longer?

4. Have a conclusion section at the end.

Author Response

Dear Reviewer:

Reviewer 2 Report

Dear Author,

It was well written article but need to be extensive english grammer checking... other than this write in details in result and discussion 

I will recommended major revision

1.      Result and discussion need to be citation in reference their work…. Such wound healing process

2.      Need  study analysis such AFM or SEM image to clarify their work

3.      Figure not clear to me

4.      Some confusion in Functional regeneration of damaged skin tissue section….

Author Response

Dear Reviewer:

Round 2

Reviewer 2 Report

i very pleased with the revisied version of manuscript and i will recommed the acceptance of manuscript for publication in present form